# Novel Cellular Therapies for Hepatocellular Carcinoma

**DOI:** 10.3390/cancers14030504

**Published:** 2022-01-20

**Authors:** Harriet Roddy, Tim Meyer, Claire Roddie

**Affiliations:** 1UCL Cancer Institute, London WC1E 6DD, UK; harriet.roddy.19@ucl.ac.uk (H.R.); t.meyer@ucl.ac.uk (T.M.); 2University College London Hospitals NHS Foundation Trust, London NW1 2BU, UK; 3Royal Free Hospital, Pond Street, London NW3 2QG, UK

**Keywords:** hepatocellular carcinoma, immunotherapy, chimeric antigen receptor (CAR) T-cells, engineered TCR T-cells

## Abstract

**Simple Summary:**

Hepatocellular carcinoma (HCC) is one of the leading causes of cancer related death worldwide. Most patients present with incurable disease and the prognosis remains poor. Immunotherapy using immune checkpoint inhibitors has become standard of care for advanced disease but only a minority respond, and novel approaches are needed. Cell based immune therapies have proven efficacy in haematological cancers and are currently being evaluated in solid tumours including HCC. Examples include Chimeric Antigen Receptor T cells (CAR-T) and T-cell receptor engineered T-cells (TCR-T). Here, we review the landscape of cellular immunotherapy for HCC, and outline how advanced engineering solutions may further enhance this therapeutic approach.

**Abstract:**

Hepatocellular carcinoma (HCC) is one of the leading causes of cancer related death worldwide. Most patients present with advanced disease, and current gold-standard management using tyrosine kinase inhibitors or immune checkpoint inhibitors (ICIs) offers modest clinical benefit. Cellular immune therapies targeting HCC are currently being tested in the laboratory and in clinical trials. Here, we review the landscape of cellular immunotherapy for HCC, defining antigenic targets, outlining the range of cell therapy products being applied in HCC (such as CAR-T and TCR-T), and exploring how advanced engineering solutions may further enhance this therapeutic approach.

## 1. Introduction

Hepatocellular carcinoma (HCC) accounts for 80% of all primary liver cancers and is a leading cause of cancer related mortality [1]. Common associations include Hepatitis B, Hepatitis C, non-alcoholic fatty liver disease and alcohol [2].

In the healthy liver, phagocytic, innate and diverse immunomodulatory CD4+ and cytotoxic CD8+ immune cell compartments permit detection and clearance of gut-derived pathogens and toxins, whilst maintaining immune tolerance [3]. Hepatotropic viruses, toxins and drug-induced necroinflammation drive proinflammatory cytokine secretion (interleukin 2 (IL-2), interleukin 7 (IL-7), interferon-gamma (IFNγ), which impairs homeostatic balance. Compensatory anti-inflammatory cytokine secretion (IL-10, IL-13 and transforming growth factor-beta (TGF-b) follows, creating a cycle of chronic damage and repair which can cause permanent structural damage, fibrosis, cirrhosis, and a predisposition towards tumorigenesis [2].

The HCC tumour immune microenvironment (TME) is diverse and complex, comprising Kupffer cells, liver sinusoidal endothelial cells, stellate cells, cancer associated fibroblasts (CAF) and dysfunctional adaptive and innate immune compartments. Research suggests that HCC is immunogenic: T-cell infiltration is associated with lower recurrence and higher 5-year OS [4], and low CD4:CD8 ratios are associated with lower tumour grade and lower recurrence rates [5].

Current treatments for early stage HCC include tumour resection, liver transplantation or ablative therapy and intermediate stage disease is managed with intra-arterial therapy [6]. However, patients with more extensive liver disease, portal vein involvement and/or extrahepatic disease require systemic therapy. The protein tyrosine kinase inhibitor (TKI) sorafenib was the first targeted agent to be approved by the US Food and Drug Administration (FDA) for advanced HCC after the SHARP trial demonstrated improved overall survival (OS) from 7.9 with placebo to 10.7 months with sorafenib [7].

Next-generation TKIs (Lenvatinib [8], regorafenib [9] and cabozantinib [10]), and the vascular endothelial growth factor (VEGF) receptor 2 monoclonal antibody (mAb) ramucirumab [11] have demonstrated similar modest improvements in clinical outcome compared with controls.

Immune checkpoint inhibitors (ICIs) are the most extensively tested immune therapy for HCC [12]. The Checkpoint040 trial (NCT01658878) of the Programmed Death Receptor-1 (PD-1) blocking antibody, Nivolumab, in advanced HCC, led to FDA approval in 2017 [13]. Objective response rates in the dose expansion phase were 20%, and Nivolumab had a manageable safety profile. Most recently, the combination of Atezolizumab (PD-L1 inhibitor) and bevacizumab (VEGF inhibitor) was demonstrated to have superior survival compared to sorafenib and has now become the first-line standard of care [14]. Recent international guidelines define the recommended first-and second line sequences based on available evidence [15]. Other novel ICIs targeting CTLA4 [16], TIM-3 and LAG-3 are also being tested [17,18,19,20,21]. ICIs are attractive for HCC as they are often well characterized agents that can slow tumour growth/ induce regression in a proportion of cases. Disadvantages include the apparent absence of curative potential, the need for ongoing treatment to maintain clinical effect (with attendant cost/health economic implications), and the risk of immune mediated adverse events (iMAEs) [22].

Adoptive cellular therapy (ACT) using both gene-modified and non-gene modified T-cells as a therapeutic modality in HCC is gaining traction. Redirecting T cells to recognize HCC tumour-associated antigens is now possible using gene engineering approaches to induce programmed expression of synthetic cell surface receptors on T cells. Examples include chimeric antigen receptor T-cells (CAR-T), and T-cell receptor modified T-cells (TCR-T) therapies, which are now being tested in clinical trials. Here, we provide an overview of current T cell therapy paradigms for HCC, and future directions for this novel therapeutic class. Figure 1 illustrates the wide range of immune-based therapies being applied in HCC, delineating what is currently approved and what is currently under investigation in clinical trials.

## 2. Non-Gene Modified Adoptive Cell Therapies

### 2.1. Cytokine Induced Killer (CIK) Cells

#### 2.1.1. CIK Biology and Background

CIKs are CD3 + CD56+ natural killer(NK)-like T-cells expanded from peripheral blood mononuclear cells (PBMC) or cord blood by expansion with anti-CD3 monoclonal antibody(mAb), IL-2, interleukin-1-alpha (IL-1α) and IFNγ [23,24]. Preclinical models demonstrate that CIKs possess non-MHC-restricted cytotoxic and proliferative activity [25,26]. A single allogeneic CIK manufacture could potentially generate multiple products for multiple patients [27].

#### 2.1.2. CIKs for HCC

Phase I clinical studies of ‘adjuvant’ CIKs in Barcelona Clinic Liver Cancer (BCLC) stage A/B disease post-surgery or trans-arterial chemo-embolization, suggest that CIKs can reduce tumour recurrence without serious adverse events or graft-versus-host-disease (GvHD) [28,29]. Further, a clinical trial of autologous CIKs was conducted in 13 HBV-associated HCC patients: 2/13 had early stage disease and 11/13 had advanced disease. CIKs were well-tolerated, with reductions in HBV viral load and slowed tumour growth in all patients, including partial response (PR) in 3/13 [30]. A subsequent Phase I basket trial of CIKs for advanced stage, refractory renal cell carcinoma, HCC and lymphoma treated 12 patients with 3 cycles of CIK therapy, infused at intervals of 3 weeks, at a median dose of 28 × 10^9^ total cells (range, 6–61 × 10^9^). CIKs were well-tolerated and 3/12 patients achieved complete response (CR) and 2/12 maintained stable disease (SD), with a mean follow up of 33 months (range, 9–44 months) [31].

#### 2.1.3. Combinatorial Approaches: CIKs + TACE

Hao et al. evaluated CIK therapy combined with transcatheter arterial chemoembolization (TACE) in patients with unresectable HCC [32]. 72 patients received CIK combined with TACE, and 74 patients received TACE alone. The 6-month, 1-year and 2-year PFS was reported as 72.2%, 40.4% and 25.3% in the CIK + TACE group, compared with 34.8%, 7.7% and 2.6% in TACE-alone group. Median OS was 31 months (95% CI, 27–35 months) in the CIK + TACE group and 10 months (95% CI 7–13 months) in the TACE-alone group. These results speak to the potential for combinatorial immunotherapeutic approaches for HCC.

In a further clinical trial of 64 HCC patients [33], TACE was sequentially combined with radiofrequency ablation (RFA), with or without prior CIK therapy (33/64 patients + CIK, 31/63 patients-CIK). CIKs were administered in 8 doses at four weekly intervals via peripheral vein or hepatic artery. At 1 year, 29/33 (88%) patients in the CIK-treated group and 23/31 (68%) in the control group were recurrence-free [33].

#### 2.1.4. Combinatorial Approaches: CIKs + Dendritic cells (DCs)

DCs can promote CIK activation, and combined DC-CIK therapy has been tested in an oncology basket study [34]. DC-CIK alone was reported to control tumour growth and improve OS [34,35]. In combination with Sorafenib, DC-CIK showed activity against the HCC target cell line BEL27402 in preclinical models, with improved cytotoxicity compared with DC-CIK alone, sorafenib alone and CIK cells alone (*p* < 0.01) [36]. DC-CIK may be further enhanced by the pro-inflammatory cytokine interleukin 24 (IL-24) [37]. IL-24-engineered DCs in co-culture with CIKs deliver significantly higher lytic activity against HCC cell lines than non-modified DCs [37].

#### 2.1.5. Future Directions for CIK Therapy

Additional clinical data will help to elucidate the potential of unmanipulated CIKs for advanced HCC. To date, combination approaches using conventional agents (e.g., Sorafenib) and experimental agents (e.g., gene-modified DC) are yielding interesting results and require further investigation.

### 2.2. Tumour Infiltrating Lymphocytes (TILs)

#### 2.2.1. TIL Biology and Background

TILs are polyclonal, tumour-targeting T-cells, expanded from patient tumour biopsies ex vivo for use as autologous therapeutics. In immunogenic tumours such as metastatic melanoma, overall response rates (ORR) are reported as 49–72%, and CR is observed in 10–20%, with durable responses in 40% [38,39,40]. Protracted ex vivo processing can lead to terminally differentiated TIL products in which immunosuppressive Treg compartments predominate. To overcome this, researchers can now selectively expand tumour-reactive/ neoantigen-responsive cytotoxic T-cells from within bulk TILs [41], and clinical studies of these products are underway in non-small cell lung cancer (NSCLC) and melanoma (NCT04032847; NCT03997474).

#### 2.2.2. TILs for HCC

In preclinical models, HCC TILs are phenotypically exhausted, expressing high levels of TIM-3 and LAG-3, with a reduced capacity for cytokine secretion [42,43]. Co-blockade of T cell immunoreceptor with Ig and ITIM domains (TIGIT) and PD1 has been shown to improve proliferation and cytokine secretion in preclinical models [43].

To date, there is a paucity of published clinical trial data on TILs for high-risk, recurrent HCC, with only a single active trial of autologous TILs (NCT04538313). Further data are warranted to better characterise TIL therapy in HCC.

## 3. Gene Modified Adoptive Cell Therapies

Successful immunotherapeutic strategies for cancer rely on the ability of the immune system to recognize tumour associated antigens (TAAs) to deliver effective anti-tumour responses. Fundamentally, the immune system is designed to promote tolerance to self, and given that all tumours arise from self-tissue, the ability to generate a potent, endogenous anti-tumour response is a significant hurdle. To overcome this, gene engineering approaches to modify immune cells with synthetic receptors such as CAR-T and TCR-T, to increase the visibility of tumour-specific antigens (TSA) and TAA, holds great promise.

### 3.1. Chimeric Antigen Receptor T-Cells (CAR-T)

#### 3.1.1. CAR-T Biology, Background, Targets

CARs are synthetic cell surface receptors engineered into immune cells to ‘reprogramme’ them to recognise and kill cells expressing specific tumour-associated targets in an MHC-independent manner [44,45]. CAR-T design is critical to function (Figure 2a). CAR structures typically comprise an extracellular, antibody-derived single-chain variable fragment (scFv) antigen-binding domain, coupled to a spacer and transmembrane domain, fused to an intracellular signalling endodomain which incorporates (at a minimum) CD3ζ signalling capability [44]. CARs with isolated CD3ζ signalling, i.e., first-generation CARs, have largely been superseded by CAR designs incorporating co-stimulatory domains such as CD28 or 41BB which have been shown to improve CAR-T expansion and cytotoxicity in vitro and in vivo [46,47]. Second generation CD19-targeting CARs with 41BB [48] and CD28 [49] costimulatory endodomains have emerged as the most impactful therapeutic paradigm shift in malignant haematology in the last decade, leading to global licensing of autologous CD19-targeting CAR-T for relapsed and refractory (r/r) B-cell cancers. Of note, third generation CARs, which incorporate two intracellular signalling domains (e.g., 41BB and CD28), are also currently under investigation [50,51,52,53,54,55,56] (Figure 2b). CAR-T for HCC is an area of active research with several promising CAR-T tumour targets identified. Here, we will focus on Glypican-3 (GPC3), alpha-fetoprotein (AFP), c-MET, Mucin 1 and NK group 2 member D ligand (NKG2DL). At time of review, 8 phase I/II CAR-T clinical trials for HCC are recruiting (Table 1).

#### 3.1.2. Glypican-3 (GPC3) Biology

GPC3 is a 65kD, glycosylphosphatidylinositol (GPI) anchored 580 amino acid heparan sulphate proteoglycan [57,58], found on the placenta, where it plays a role in morphogenesis (via activation of the Wnt pathway), and in several solid tumours, including 72% of cases of HCC, where it portends to poor prognosis [59,60]. Critically, GPC3 is minimally expressed on other tissues including normal and cirrhotic liver [61,62]. Soluble GPC3 is found in 53% of HCC patients, and is under development as a disease biomarker [59]. GPC3 appears to have a role in both HCC prognosis and disease development [63,64,65]. SiRNA-mediated GPC3 silencing in primary HCC cells leads to reduced proliferation, increased apoptosis and impaired tumour cell migration compared to controls [66].

#### 3.1.3. Glypican-3 (GPC3) Targeting CAR-T (Preclinical)

2nd and 3rd-generation GPC3-targeting CAR-T (GPC3-CAR-T) possess activity against HCC cell lines (HepG2, Huh-7) in vitro and in vivo [63,67,68]. Evaluation of 3rd generation GPC3-CAR-T in a patient derived xenograft (PDX) HCC model demonstrated a reduction in tumour burden compared with CD19-CAR-T controls. The authors report a single mouse with resistance to GPC3-CAR-T, in which the PDX had high PDL1 expression. To overcome this, they suggest that CAR plus ICI may be valuable in HCC [68], as has been shown in CAR-T models of breast cancer (Her2) and mesothelioma (mesothelin) [69,70].

#### 3.1.4. Glypican-3 (GPC3) Targeting CAR-T (Clinical)

2 sequential phase I trials of autologous 2nd generation 41BBζ GPC3-CAR-T in adults with r/r advanced HCC (NCT 02395250; NCT03146234) are reported [71]. The first investigated GPC3-CAR-T in the context of 2 lymphodepletion regimens: cyclophosphamide (Cy) alone, at a dose of 500–1000 mg/m^2^/day for 1–2 days, versus Cy at a dose of 500 mg/m^2^/day for 1–3 days, in combination with fludarabine (Flu) at a dose of 25–30 mg/m^2^/day for 2–4 days. In the second study, all patients received Cy (500 mg/m^2^/day for 1–2 days) and Flu (20–25 mg/m^2^/day for 3–4 days) in combination.

In the first study, GPC3-CAR-T was administered as a split dose in 8 patients, all of whom received an initial dose of 1 × 10^5^ CAR-T/kg. Repeated, incremental doses were administered as 6 infusions over 13 days to a maximum dose of 2 × 10^9^ CAR-T/kg, with the aim of defining the maximum tolerated dose (MTD). In the second study, all 5 patients received the MTD recommended from the first study in one dose (20 × 10^8^ CAR-T/kg). A single patient received an additional dose of 11.1 × 10^8^ cells approximately 6 weeks following completion of initial therapy. In total, 13 patients were infused with a median of 19.9 × 10^8^ GPC3-CAR-T (7.0–92.5 × 10^8^) [71].

GPC3-CAR-T product characterization revealed enrichment for terminally differentiated, CD45RA+ re-expressing effector memory T-cells (78.2%) and effector memory T-cells (14.1%), with few central- or stem-cell memory T-cells [71]. Terminally differentiated T-cells have been shown to negatively impact CD19CAR-T activity, and this is likely also to be the case in HCC [72,73]. Research to optimize leukapheresis and product profile may enhance GPC3-CAR-T functionality [74].

Any grade cytokine release syndrome (CRS) was reported in 9/13 (69%) patients, with no grade ≥3 CRS events. 2/13 patients received tocilizumab (IL-6 receptor antagonist) and 4/9 received corticosteroids. 1/13 died from multi-organ failure in the context of grade 5 CRS following a total CAR-T dose of 20 × 10^8^ cells [71].

1/13 patients achieved PR, 2/13 maintained SD and 8/13 developed PD [71]. Median OS for the whole cohort was 42.0% and 10.5% at 1 and 3 years, respectively. Whilst these response rates are disappointing, the toxicity profile was broadly tolerable, and this study has provided the foundation for further development of ‘optimised’ GPC3-CAR-T approaches. To this end, ‘armoured’ GPC3-CAR-T designs, with engineered modules for 41BBL and IL-15/IL-21 (NCT02932956) [75], combinatorial approaches using concurrent TKIs or IPIs (NCT03980288), and targeted delivery via hepatic artery infusion (NCT03993743) are all under investigation.

The potential impact of soluble GPC3 on GPC3-CAR-T function is not well characterised, but it is possible that soluble GPC3 may block access to cell-surface GPC3 [76]. It will be important for future studies to consider this in the design and preclinical evaluation of next-generation GPC3-CAR-T.

#### 3.1.5. Alpha Fetoprotein (AFP) Biology

AFP is a 591 amino acid glycoprotein, and is expressed almost exclusively during embryonic development, with the exceptions of chronic liver inflammation [77] and 60–80% of HCC, where it is found both intracellularly (processed to peptide) and on the cell surface, presented on MHC class I. AFP is also secreted in the serum and is used as an HCC biomarker. Biologically, AFP promotes tumour proliferation and portends to poor prognosis [78,79].

#### 3.1.6. Alpha Fetoprotein (AFP) Targeting CAR-T (Preclinical)

Due to the intracellular localisation of the antigen and its expression on MHC, CAR-T approaches are challenging. One group developed 2nd generation AFP-CAR-T(CD28/CD3ζ) using ET1402L1, a fully humanised binder specifically targeting the AFP_158–166_ epitope presented by HLA-A*02:01 [80]. HLA-A*02:01 CAR-T has the disadvantage of restricting patient access to specific HLA-matched groups yet provides an opportunity for the CAR-T field to target intracellular antigens, which has not been previously possible. This is critical in solid cancers where there is a paucity of cell surface targets, to broaden the therapeutic scope of CAR-T.

AFP-CAR-T in vitro is demonstrated to be cytotoxic towards HEPG2 and SK-HEP-1-MG HCC cell lines. To address concerns that circulating AFP can be cross-presented on MHC Class I by non-HCC cells, AFP-CAR-T was tested against a range of A02+/AFP− cell lines, cultured in recombinant human AFP (0.4 μg/mL), but no cytotoxicity was observed.

In preclinical models, intra-tumoral injection (and to a significantly lesser extent, IV injection) of AFP-CAR-T in NSG mice bearing HLA-A*02:01 HCC SK-HEP-1-MG and HEPG2 tumour cells expressing the AFP_158–166_ epitope, controlled tumour growth compared to controls [80].

#### 3.1.7. Alpha Fetoprotein (AFP) Targeting CAR-T (Clinical)

A phase I clinical trial of AFP-CAR-T in AFP+ HCC via IV and intra-hepatic artery dosing was terminated, with no outcomes reported to date (NCT03349255).

#### 3.1.8. c-MET Biology and Background

c-MET is a proto-oncogene encoding a cell surface tyrosine kinase receptor with high affinity for hepatocyte growth factor (HGF), acting to promote cell growth, invasion and resistance to apoptosis [81]. Aberrant c-Met activity occurs in 50% of HCC cases, and is associated with rapid tumour growth, invasive disease, and poor prognosis [82]. The c-MET inhibitor Tivantinib has been tested in advanced HCC, but without impact on OS or EFS [83].

#### 3.1.9. c-MET Targeting CAR-T (Preclinical)

Bispecific CAR-T targeting c-MET + PD-L1 demonstrates selective cytotoxicity against HCC cell lines in vitro and in vivo more effectively than monovalent c-MET-CAR-T. In vivo, HEPG2 xenograft models received bispecific or c-MET-CAR-T, and tumour burden was reduced (but not eliminated) in all mice by the end of the experiment (day 13) compared to control CAR [84]. To date, this approach has not reached clinical translation.

#### 3.1.10. Mucin 1 Glycoprotein 1 (MUC1) Biology and Background

MUC1 is a membrane-bound glycosylated phosphoprotein, expression of which is induced by inflammatory cytokines (Figure 1). MUC1 is overexpressed and/or qualitatively altered in HCC [85,86,87] and has been validated as a potential HCC target by immunohistochemical analysis, with strong positivity demonstrated in 70.8% of cases (with no expression in normal liver tissue) [88]. MUC1 strongly correlates with HCC prognosis and metastasis: 67.7% of MUC1+ patients develop metastasis at 3 years compared with 31% of patients without MUC1 expression [88].

#### 3.1.11. MUC1 Targeting CAR-T (Preclinical)

Proof-of-concept for MUC1 as an effective CAR-T target is evident in breast cancer, where TAB004 (mAb targeting glycosylated MUC1), engineered into CAR-T format, has shown target-specific cytotoxicity, cytokine secretion and reduced tumour burden in a xenograft model [89]. In vitro studies show MUC1-specific cytotoxicity against HCC cell lines [90].

#### 3.1.12. MUC1 Targeting CAR-T (Clinical)

A clinical trial of MUC1-CAR-T is underway in multiple solid tumours including HCC (NCT02587689), but outcomes have not yet been reported. A fourth generation MUC1-IL-22-secreting CAR-T product for head and neck squamous cell carcinoma has been developed for clinical trials [91].

#### 3.1.13. NK Group 2 Member D Ligand (NKG2DL) Biology and Background

NKG2DL is a type II transmembrane protein expressed on NK and T-cells with a role in NK cell activation, and provision of co-stimulation to CD8+ and CD4+ T-cells. NKG2DL is overexpressed in HCC (but not normal liver), highlighting its potential as an immunotherapeutic target [92]. Targeting specific ligands for NKG2D such as MICA, which is not expressed in most normal tissues and is localised in the cytoplasm of epithelial cells, offers of a potentially safe target for NKG2DL-based CAR-T [93]. Concerns regarding off-tumour NK (and T-cell) depletion will require full evaluation [92].

#### 3.1.14. NKG2DL Targeting CAR-T (Preclinical)

In vitro testing of 3rd generation NKG2DL-CAR-T demonstrated specific cytotoxicity towards SMMC-7721 and MHCC97H cell lines but not Hep3B cells (NKG2DL-negative). Cytotoxicity was NKG2DL dependent: NKG2DL low MICA and ULBP2 cells were not susceptible, but NKG2DL high HEP3B were susceptible. In vivo, NKG2DL-CAR-T were effective in NOD-Prkdc^scid^//Il2rg^tm1^/Bcgen mice (B-NDG) with SMMC-7721 subcutaneous tumours: 50% (3/6) of treated mice achieved CR and 50% (3/6) achieved PR at 19 days post-infusion.

#### 3.1.15. NKG2DL Targeting CAR-T (Clinical)

A phase I clinical trial of NKR-2 (NCT02203825) in 7 patients with acute myeloid leukaemia and 5 patients with multiple myeloma reported no obvious toxicities or other adverse events. Full clinical data are pending from subsequent dose escalation [94]. To date there are no data in HCC.

#### 3.1.16. Other Potential CAR-T Targets for HCC

A variety of other potential HCC targets are currently under evaluation [95,96,97,98]. CD147 is a transmembrane glycoprotein that is highly expressed in HCC [99] with a role in proliferation, invasion and metastasis and as a potential biomarker [100]. As a safety system, Zhang et al. developed a ‘tet-on’ inducible system for anti-CD147-CAR-T, with a requirement for the administration of doxycycline for CAR activity. Omission of doxycycline reduced CD147-HepG2 target cell lysis in vitro cytotoxicity assays from 50% to 10%. In vivo, using the subcutaneous Huh-7 NSG mouse model, CD147CAR + doxycycline led to a significant reduction in tumour volume when compared to CD147CAR alone [101]. A phase I clinical trial is currently underway (NCT03993743).

The MAGE (melanoma antigen gene family)-1 family comprises 12 members that are expressed almost exclusively in cancer tissues. MAGE-1 and -3 mRNA expression is observed in 68% of HCC tumours [102]. Functionally, aberrant MAGE expression can lead to altered signalling and tumuorigenesis [103]. To date, there is limited evidence for MAGE-CAR-T in HCC.

NY-ESO-1 (New York esophageal squamous cell carcinoma), is a cancer testes antigen expressed in spermatocytes, oogonia, the placenta and cancers including 43.9% of cases of HCC [104,105]: At present, only engineered TCR-T approaches targeting NY-ESO-1 have been trialled in HCC (NCT01967823).

### 3.2. T-Cell Receptor Transduced T Cells (TCR-T)

#### 3.2.1. TCR-T Biology, Background, Targets

Synthetic, engineered TCRs (Figure 2c) are designed to specifically recognise intracellular TAA/TSA following processing and presentation of short antigenic peptide fragments on HLA class I and II molecules [106]. A key advantage of TCR-T is the ability to target and bind intracellular antigens expressed on HLA, even at low target density [107]. Disadvantages include the requirement for HLA matching, limiting this therapy to a minority of mostly HLA-A*0201 patients [108], and the engineering complexity of generating high affinity, high avidity synthetic αβ TCR for TAA-specific targeting, avoiding the common hurdles of TCR chain mispairing and low TCR expression [109]. At low antigen density, there is also a risk that expression overlaps between normal and cancer tissue. Tuning TCR affinity using physiochemical and in silico methods could help advance TCR-T binder development to enable differentiation between non-malignant and cancer cells [77]. In this section we focus on AFP and viral associated peptides as TCR-T HCC tumour targets. At time of review, 7 phase I/II TCR-T clinical trials for HCC are recruiting (Table 2) [110]. There are several additional studies on clinicaltrials.gov with ‘status unknown’ or ‘not yet recruiting’ which focus on patients with HCC recurrence post liver transplantation (NCT02686372, NCT04677088, NCT02719782).

#### 3.2.2. AFP TCR-T Biology and Background

AFP is processed and presented on HLA and represents a good target for TCR-T approaches, with clinical trials underway using the well described AFP_158–166_ epitope (NCT03132792) [111]. Butterfield et al. describe 3 novel epitopes (AFP_542–550_; AFP_137–145_, and AFP_325–354_) expressed on HLA class I for which in vitro validation demonstrates cytotoxicity and IFNγ secretion towards HLA-A*0201+/AFP+ tumour cells. Epitope immunogenicity was tested in a transgenic mouse model [112]. Together, this evidence confirms that there are four naturally occurring, immunogenic AFP epitopes that represent potential targets for TCR-T.

#### 3.2.3. AFP TCR-T (Clinical)

In 2019, initial results were presented for a first-in-human study of affinity-enhanced autologous SPEAR T-cells targeting AFP (NCT 03132792) [113]. Patients with HLA-A*02:01 or HLA-A*02:642 haplotypes with positive immunohistochemical staining for AFP (≥1+ in ≥20% of HCC cells) or positive serum AFP (≥400 ng/mL) were eligible. In cohort 1, with the aim of determining MTD, SPEAR T-cell doses ranging from 100–1000 million total cells were administered following Flu/Cy. 5/5 infused patients achieved SD as their best response. MTD for Cohort 3 was defined as 5 ×10^9^ SPEAR T-cells (range, 5.0–5.6 × 10^9^). In Cohort 3, 2 of 3 patients developed PD, and 1 of 3 patients achieved PR, ongoing at week 8 and treatment was well-tolerated. Full results have not yet been published, as long-term follow up is ongoing [111].

#### 3.2.4. Viral Associated Peptides as TCR-T Targets

HBV accounts for 80% of HCC cases in Asia, through chronic inflammation and viral integration into the hepatocyte genome [114]. Targeting viral, non-self-antigens is attractive for TCR-T therapy as the risk of on-target, off-tumour toxicity is reduced, and even low-level target expression lends itself to TCR-T targeting. HBV infection generally induces high affinity TCRs which potentially represent a potent therapeutic, with the ability to lyse HBV-infected cells. However, the major drawback is that in HBV-associated HCC, viral antigens are likely to be expressed on non-malignant liver tissue, leading to a risk of severe liver damage [115].

Gehring et al. described a liver transplant patient with extrahepatic HCC in which HBV-DNA integration produced only surface HBV (HBsAg), while HBV-DNA was absent in the blood and the transplanted liver [115,116]. The HCC cells presented HLA-A0201/HBV peptide complexes, with homogeneous expression across the tumour. A specific TCR targeting the HLA-A-*0201/HBs183-91 complex was identified and cloned to generate autologous, patient HBV-TCR-T cells. The patient received a single dose of 1.2 × 10^4^ HBV-TCR-T/kg without lymphodepletion. These cells proliferated and reduced HBsAg levels in the blood without damage to normal tissue [115]. One key question arising from this case is whether HBV-TCR-T could be used more widely for HCC patients with relapse post-liver-transplant, as relapse rates are high (50% within 5 years) [117] and serum HBsAg is associated with recurrence [118]. TCR-T could potentially be used as prophylaxis to prevent relapse in HBsAg+ patients, post-transplant.

## 4. What Is the Future of T cell therapy for HCC?

To date, the HCC cellular immunotherapy field, summarized in Table 3, is in its infancy and significant challenges need to be addressed in order to bring these therapies to the forefront of HCC management (Figure 3).

### 4.1. CAR-T: Enhancing Tumour Recognition, Overcoming Escape and Reducing Toxicity

Major hurdles for targeted T cell therapies for HCC include the paucity of true TSAs, the heterogeneous expression of TAAs across tumours, and the promiscuity of expression of many solid tumour targets on normal tissues, conferring a risk of on-target, off-tumour toxicity. In the CAR-T space, targeting multiple HCC antigens may enhance specificity and potentially be exploited to improve safety. This may be achieved by infusing two different CAR products targeting different TAAs; by the expression of two different CAR scFv within one vector (Tandem CAR-T or bispecific CAR-T), or by the use of logic-gated CAR-T [119]. Preclinical evaluation of logic-gated GPC3-synNotch-inducible CD147-CAR-T cells to target HCC demonstrated selective cytotoxicity towards dual antigen positive targets without on-target/off-tumour toxicity in a human CD147 transgenic mouse model [120].

Another approach to address the on-target, off-tumour risk of CAR-T in solid cancers is the use of affinity-tuned CAR-T, where the affinity of the CAR scFv can be aligned with antigenic expression on the tumour (but not normal tissue). This has been tested in an ErbB2 [121] model, where low affinity CAR-T was designed to recognise only high-expressor malignant cells, but was insensitive to cells with normal physiologic antigen expression.

To further enhance safety, suicide genes, and tunable constructs (where CAR-T signaling can be modulated using drugs), can be incorporated into CAR designs to allow CAR-T deletion/impedance in the event of toxicity [122].

### 4.2. CAR-T: Enhancing Trafficking and Persistence

The development of bolt-on modules and strategies to enhance CAR-T trafficking and access to the HCC tumour site is an area of active investigation. HCC creates a physical barrier to T-cell trafficking, with a well-established fibrotic extracellular matrix (ECM) evolved during prior chronic inflammation [2]. Cultured T-cells lack heparinase, which is critical for degradation of ECM [123] and several groups are currently investigating whether CAR-T cells designed to degrade the ECM may have a trafficking advantage in HCC. CAFs (cancer associated fibroblasts) express fibroblast activation protein which is upregulated on HCC in the hypoxic TME [124]. FAP-directed CAR-T may have a role in HCC, targeting the fibrotic HCC TME [125]. Other approaches to enhance tumour trafficking include intrahepatic CAR-T infusion. This also has the potential to reduce the cell dose required for clinical effect, and to reduce potential side effects by limiting systemic exposure [126].

Chemokines play a key role in leukocyte migration/homing and cancer migration/progression [127,128]. By engineering expression of chemokine receptors into CAR-T products, it may be possible to exploit tumour-associated chemokine gradients to enhance CAR-T tumour trafficking. CXCL2 is highly expressed in HCC [129], and engineered overexpression of the CXCR2L receptor (CXCR2) on GPC3-CAR-T cells was shown to enhance CAR-T infiltration and expansion at the tumour site in a xenogeneic HCC model [129].

In preclinical models, CAR-T cells that co-express proinflammatory cytokines have been associated with superior proliferation, persistence and better tumour control [130]. GPC3-CAR co-expressing IL-21 and IL-15 have demonstrated superior in vitro and in vivo expansion, persistence and anti-tumour activity [131]. A clinical trial of GPC3 CAR-T cells incorporating IL-15 and IL-21 modules for HCC is recruiting (NCT04093648).

### 4.3. CAR-T: Overcoming the Immunosuppressive Tumour Microenvironment

Modules to enhance resistance to solid tumour immune-inhibitory signals, and to recruit components of the endogenous immune system to accelerate tumour rejection are all under investigation to enhance feasibility of CAR-T for solid tumours. Previous studies have revealed that CAR-T-derived cytokines activate host macrophages and transform the TME from immunosuppressive to immunostimulatory [132,133]. In ovarian cancer, CAR-T cells that secrete IFN-γ and GM-CSF can activate tumour-associated macrophages, decrease the expression of regulatory factors, and increase IL-12 production, promoting endogenous immunity and inhibiting tumour growth. ICIs may have a role alongside ACT as a potential combinatorial approach [134]. Engineering ACT with inbuilt checkpoint blockade could provide an ‘all in one’ strategy, e.g., PD-1-deficient GPC3-CAR-T [135].

### 4.4. TCR-T: Enhancing Tumour Recognition and Reducing Off-Tumour Toxicity

Arguably the main challenges for TCR-T therapeutics are the identification of appropriate target antigens, the ability to refine and optimise TCR-T binding affinity, and the avoidance of on-target, off-tumour toxicity.

Identification of neo-antigen specific TCRs is a key hurdle to developing novel TCR-T therapies. High-throughput sequencing of the immune repertoire in oncology (HTS-IR) technology at bulk and single cell levels (including computational methods like TraCeR and single cell TCRseq for reconstructing TCRs and identifying immunogenic neoantigens) have proved to be useful tools for the analysis of diversity, dynamics and clonality of T cells [136]. With cost savings from advances in sequencing and bioinformatics technology [137], these platforms may find more widespread use in target identification and validation, at a speed that outperforms conventional laboratory-based methods.

Medigene has developed a humanized mouse model/platform for identification and validation of affinity-enhanced TCRs [138]. Affinity-enhanced TCRs may confer better anti-tumour response to TSA/TAAs, but require careful evaluation for cross-reactivity with self-derived peptides to determine whether there is an optimal affinity window for enhanced function without non-specific activation [139].

Strategies to reduce the risk of off-target TCR-T toxicity include intra-tumoural TCR-T administration, to minimise systemic exposure [140]. Engineered solutions include tumour-specific TCR-T expressed in tandem with immune-inhibitory CAR-T, the latter designed with an scFv targeting an antigen on normal tissue, fused to an inhibitory cytoplasmic domain, such as PD-1 [141]. This may enable TCR-T cells to distinguish between tumour cells and normal cells using the same antigen. Suicide switches such as inducible caspase 9 (iCas9) can be incorporated into the TCR-T cassette to permit apoptosis induction using a small molecule in the event of toxicity [142].

## 5. Conclusions

ACT for solid tumours is in its infancy and is challenging for a variety of reasons, outlined in this review. HCC remains a strong candidate for cellular immunotherapy, due to its underlying immune and inflammatory pathogenesis, and progress to date in the field has been very encouraging. Drawing from learnings in the blood cancer space and with a deeper understanding of HCC pathophysiology, current investigational T cell therapies may be improved by modifications to enhance HCC targeting, trafficking and immune resistance, towards improved safety and clinical outcomes. To this end, further preclinical refinements and patient access to early phase clinical trials is key.

Currently, cell therapies for HCC are at early stage of development and none have yet been approved. Their position in the therapeutic algorithm will be determined by the magnitude of benefit demonstrated in large clinical trials. In the future, it is possible that novel predictive biomarkers will inform treatment selection and sequencing of drugs for HCC. Presently, the only biomarkers for eligibility in CAR-T and TCR-T trials are target antigen expression and (for TCR-T only) HLA expression analysis. It is too early to predict whether biomarkers for response and toxicity will be identified using ACT for HCC, but current clinical trials will be informative in this regard.

To date, approved drugs for advanced HCC have been evaluated in Child Pugh A liver disease in patients with preserved liver function. We strongly recommend that clinical trials of cellular therapies in HCC apply the same criteria, such that potential hepatotoxicity associated with these novel therapies can be appropriately defined and managed.

Ultimately, rational combinations and/or sequencing of T-cell therapies with small molecules and CPI may be required for optimal clinical outcomes in advanced HCC and should be explored in well-designed clinical studies.

## Figures and Tables

**Figure 1 cancers-14-00504-f001:**
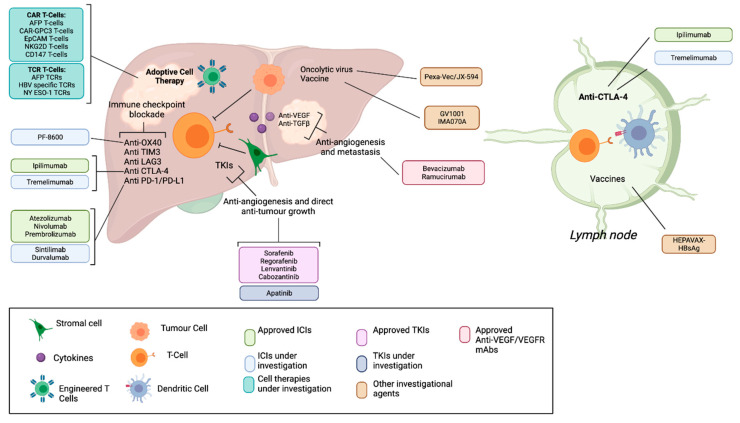
Immune-based and TKI therapies for HCC. Approved drugs and investigational agents are listed, including checkpoint inhibitors (ICIs), adoptive T cell therapies, small molecule inhibitors of angiogenesis and tumour growth, monoclonal antibodies targeting angiogenesis and oncolytic viruses and vaccines.

**Figure 2 cancers-14-00504-f002:**
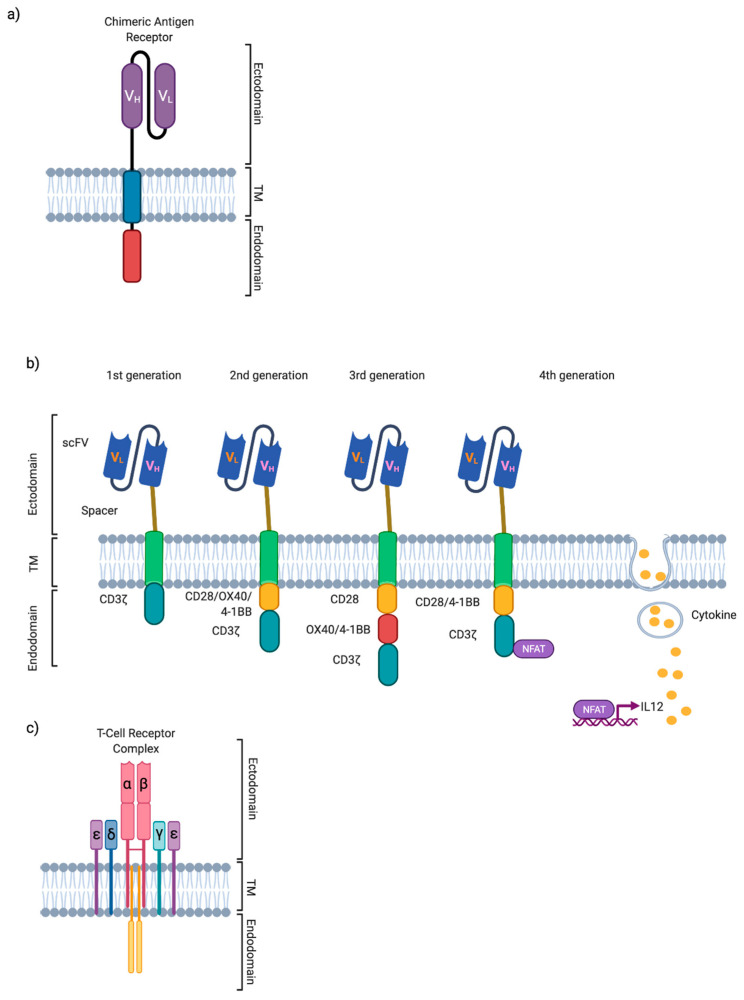
Structure of T-cell receptors (TCR) and Chimeric antigen receptors (CAR-T). (**a**) CAR-T cells recognise tumour antigens in an MHC-unrestricted manner. Target binding induces T cell activation, proliferation and cytotoxicity. (**b**) CAR-T design has evolved over time. 1st generation CARs comprise an extracellular scFv recognising a specific antigen connected to a transmembrane domain via a hinge with a CD3ζ endodomain. 2nd generation CARs include an additional costimulatory moeity such as CD28, whereas 3rd generation CARs have two co-stimulatory domains linked to the CD3ζ chain, e.g., 41BB and CD28, with the aim of improving CAR-T functionality and persistence. 4th generation CARs, ‘armoured CARs’, incorporate modules for cytokine production, e.g., IL-12, with the aim of improving CAR-T proliferation and persistence whilst concurrently stimulating endogenous immune compartments within the tumour microenvironment. (**c**) The T cell receptor (TCR) complex comprises engineered α and β chains in association with CD3 chains. Engineered TCRs interact with major histocompatibility complex (MHC)-antigen peptide complexes on cancer cells, to induce T cell activation and to promote tumour eradication.

**Figure 3 cancers-14-00504-f003:**
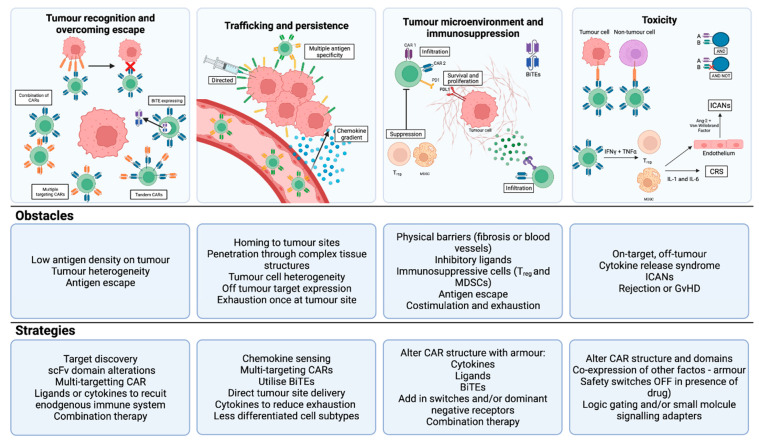
Solid tumour CAR T-cell challenges and potential strategies to overcome these. There are many obstacles to the design and delivery of effective CAR T-cell therapy for solid tumours. This includes physical barriers to tumour infiltration, heterogeneity of antigen expression, immunosuppressive factors in the tumour microenvironment and potentially challenging clinical toxicities. There are several novel engineering solutions to address each of these issues, as outlined in this figure. Key: BiTEs, bi-specific T cell engagers; CRS, cytokine release syndrome; CTLA-4, cytotoxic T lymphocyte protein 4; ICANS, immune effector cell-associated neurotoxicity syndrome; MDSCs, myeloid-derived suppressor cells; PD-1, programmed cell death 1; PD-L1, programmed cell death 1 ligand 1; scFv, single-chain Ig variable fragment; TGFβ, transforming growth factor-β; Treg, regulatory T cells.

**Table 1 cancers-14-00504-t001:** CAR for HCC—only included trials with known status, all trials are for adult cancers. Information included where available, however due to non-uniform reporting of clinical trial details there is some variation.

Study Number	Phase	Intervention	CAR-T Type	Population under Study	Inclusion/Exclusion Criteria	End Points + Expected Toxicities	Target Sample Size (n)	Status
NCT03993743	I	CD147 targeted CAR T	3rd Generation doxycycline inducible system. Autologous	Advanced HCC	Untreatable by surgery or local therapy or has postoperative progressions, has failed at least one line of standard systemic chemotherapy	DLT and MTD of CD-147 CAR-T hepatic artery infusions Activity of CAR-T cell hepatic artery infusions CAR-T detection in extrahepatic sites	34	Recruiting
NCT04121273	I	GPC3 targeted CAR T	2nd Generation Autologous	Advanced HCC	Cancer has progressed after treatment, or cannot receive standard of care.	Dose limiting toxicity such as fever or jaundice Radiological evaluation of tumour size Peripheral tumour marker Peripheral CAR-T detection by flow cytometry	20	Recruiting
NCT02905188	I	GPC3 targeted CAR T	2nd Generation Autologous Lymphodepletion regimen (Cytoxan and Fludarabine)	Previously treated HCC	Relapsed or refractory to treatment that has metastasized or cannot receive other standard lines of therapy	Dose limiting toxicity such as any grade 5 event, grade 2 or 4 allergic reaction, both haematologica and non haematologic of grade 4 that fails to return to grade 2 within 72 hurs. Grade 3 or 4 CRS. CR/PR. CAR-T persistence by PCR.	14	Recruiting
NCT03198546	I	GPC3 T2 targeted CAR T	3rd/4th Generation GPC3 and/or TGFb CAR with/without IL7/CCL19 and/or scfv against PD1/CTLA4/Tigit Autologous	Advanced HCC	Advanced disease and not eligible for alternative therapy	Dose limiting toxicity that is irreversible or life threatening; haematologic or non-haematologic grade 3–5. CR/PR Persistence measured by PCR	30	Recruiting
NCT03884751	I	GPC3 targeted CAR T	2nd Generation Autologous	Advanced HCC	Patients must not be eligible for surgery or have progressive disease after standard therapies.	Dose limiting toxicity and maximum tolerated dose Pharmacokinetics including CAR-T expansion and persistence measured by PCR	15	Recruiting
NCT03941626	I/II	EGFRviii/DR5 targeted CAR-T/TCR-T	2nd Generation Autologous Lymphodepletion regimen (Cyclophosphamide and fludarabine)	Basket trial including HCC	Multi tumour type but for liver patients must be untreatable by surgery or postoperative recurrence or no effective treatment	Adverse events and clinical response measured by change in tumour volume by CT and MRI	50	Recruiting
NCT03013712	I	EpCAM targeted CAR T	2nd Generation Autologous	Basket trial including HCC	Multi tumour type but for liver patients must be untreatable by surgery or postoperative recurrence or no effective treatment	Toxicity profile and antitumour efficacy Persistence of CAR-T cells in the blood measured by flow cytometry	60	Recruiting
NCT03980288	I	GPC3 Targeted CAR T	4th Generation Autologous Lymphodepletion regimen (Cyclophosphamide and fludarabine)	Advanced HCC	Advanced hepatocellular carcinoma and refractory or intolerant to current standard systemic treatment	Dose limiting toxicity and maximum tolerated dose CAR-T expansion measured by PCR	36	Recruiting
NCT04506983	I	GPC3-CAR T Cell	2nd Generation Autologous	Advanced HCC	Advanced disease BCLC B/C	Percentage of any adverse events Overall remission rate Proliferation of CAR T Cells	12	Not yet recruiting
NCT04550663	I	NKG2D CAR-T (KD-025)	Autologous	Basket (including HCC)	For those with advanced disease that is not eligible for other treatment. NKG2DL+	Maximum tolerated dose Adverse events occurrence monitoring Objective remission rate	10	Not yet recruiting
NCT02395250	I	GPC3 CAR T cell	2nd Generation Autologous	Previously treated HCC	Untreatable by surgery or postoperative recurrence with no effective treatment	Any adverse events incidence as a result of CAR-T cells	13	Completed
NCT02723942	I	GPC3 CAR T cell	No generation mentioned Autologous	Previously treated HCC	Untreatable by surgery or postoperative recurrence with no effective treatment	Effect of CAR on tumour through reduction in tumour burden, assessment via CT or PET. Safety profile related issues such as fever and jaundice.	N/A	Withdrawn
NCT04270461	I	NKG2D CAR T cell	2nd Generation Autologous	Basket NKG2DL+(including HCC)	Patients with relapsed/refractory disease	OS and safety profile Delivery via hepatic portal artery	N/A	Withdrawn
NCT04093648	I	GPC3 CAR T with IL21 and 15 (TEGAR)	4th Generation Autologous Lymphodepletion (Cytoxan and Fludarabine)	Preciously treated HCC	Untreatable by surgery or postoperative recurrence with no effective treatment	DLT including neurotoxicity and CRS Response rate (partial or complete)	N/A	Withdrawn—incorporated into another study
NCT03349255	I	ET1402L1-CAR T cell	2nd Generation Autologous	Previously treated liver cancer AFP+	No available curative therapeutic options and a poor overall prognosis.	DLT and toxicity such as fever, CRS, neutropenia Response rate CAR-T cell engraftment	3	Terminated

**Table 2 cancers-14-00504-t002:** TCR-T trials for HCC—only included trials with known status, all trials have been conducted in adults.

Study Number	Phase	Intervention	Source of Cells	Population under Study	Inclusion/ Exclusion Criteria	Trial End Points	Target Sample Size (n)	Status
NCT03132792	I	AFPC332 T Cells	Autologous	HLA-A02+ AFP+ HCC	Relapsed or refractory disease	DLTs and AEs Time intervals between infusion and response	45	Recruiting
NCT04368182	I	C-TCR055 (AFP TCR)	Autologous	HLA-A02+ AFP+ HCC	Unresectable disease, relapsed or refractory	Safety through adverse event monitoring and ORR	3	Recruiting
NCT03971747	I	C-TCR055 (AFP TCR)	Autologous	HLA-A02+ AFP+ HCC	Unresectable disease, relapsed or refractory	Safety through adverse event monitoring and ORR	9	Recruiting
NCT03634683	I/II	LioCyx	Autologous	HLA Class I HBV HCC	Recurrent HBV related HCC post transplantation	Safety by reporting of adverse events, ORR and quality of life measurements	72	Not yet recruiting
NCT04502082	I/II	ET140203 T cells (ARYA-1)	Autologous	HLA-A02+ AFP+ HCC	Advanced disease	Incidence and severity of adverse events	50	Recruiting
NCT03888859	I	ET1402L1-ARTEMISTM T cells	Autologous	HLA-A02+ AFP+ HCC	Advanced disease where patients have no available curable therapeutics	DLTs Administration route assessment; IV/intrahepatic/intratumoural	12	Completed
NCT03965546	I	ET140202 AFP T cell combination with wither TAE or Sorafenib	Autologous	HLA-A02+ AFP+ HCC	Advanced disease where patients have no available curative therapeutics	Adverse events frequency and T cell expansion	27	Recruiting
NCT03899415	I	HBV specific TCR redirected T Cell	Autologous	HLA Class I HBV HCC	Advanced HBV related HCC post hepatectomy or radiofrequency ablation	Incidence of adverse events and ORR	10	Recruiting
NCT03132792	I	AFPᶜ³³²T cells	Autologous	HLA-A02+ AFP+ HCC	Advanced, relapsed or refractory disease	DLTs and adverse event incidence and any response rate	45	Recruiting
NCT01967823	II	Anti-NY ESO-1 mTCR (murine TCR)	Autologous	HLA-A*0201 Basket (including HCC)	Advanced disease NY-ESO-1 positive tumours	Clinical response (CR/PR) and TCR expansion by PCR	11	Completed
NCT03441100	I	IMA202–101 TCR T cells targeting MAGE-1	Autologous	Basket (including HCC) HLA restricted	Advanced or metastatic tumour MAGE-1 + tumour	Adverse event incidence and persistence of T Cells	15	Recruiting

**Table 3 cancers-14-00504-t003:** Outline and comparison of the 4 main types of cellular therapies key characteristics.

Cell Therapy	Antigen(s) Recognised	MHC Restricted	Advantages	Disadvantages
TCR-T *TCR-engineered T Cells*	Peptide/MHC, intracellular targets possible	Yes	Sensitive antigen recognition Can target intracellular tumour antigens	MHC restricted, limiting patient access Single antigen targeting, prone to antigen escape Complex gene engineering/design Expensive to manufacture Bespoke, patient-specific
CAR-T *Chimeric antigen receptor T-cells*	Cell surface antigens	No	Not MHC restricted High affinity antigen recognition Strong signalling through CD3ζ Can persist in vivo over months and years	Cell surface antigen restricted Single antigen targeting, prone to antigen escape Expensive to manufacture Bespoke, patient-specific (for autologous CAR-T)
CIKs *Cytokine induced killer cells*.	Not antigen specific	No	Heterogeneous product Not antigen specific No gene engineering required	Limited clinical experience and data from trials
TILs *Tumour infiltrating lymphocytes*	Multiple tumour associated antigens	Yes	Heterogeneous product Not antigen specific No gene engineering required	Limited clinical experience and data from trials Complex manufacture How to overcome ‘exhausted’ T-cell phenotypes in TIL products Need accessible tumour tissue to make products Requirement for concurrent IL-2 to promote TIL expansion in patients

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
