# Peer review of "Novel Cellular Therapies for Hepatocellular Carcinoma"

_cancers, 2022, doi:10.3390/cancers14030504_

Round 1
Reviewer 1 Report
Dear Authors,
this is an interesting Review on Novel cellular therapies for HCC.
Gold standard therapy for advanced HCC is based on TKIs and more recently on immune check point inhibitors (Atezolizumab in combination, Nivolumab, etc), but in many countries they are not yet available or describable. These last therapies are promising and are changing the treatment panorama for advanced HCC, however they are not conclusive, some patients may not have any response and some other may lose the initial response. Clinical practice is still on going this aspect.
In this contest, novel approaches are needed and Cellular therapies can be the right direction.
As it is a very novel and specific field, I would suggest to add in this work some more general descriptions and explanations on Cellular therapy, to be better understood by practical clinicians.
The following minor modifications are suggested:
-In Simple summer or in Abstract, the sentences on the epidemiology of HCC can be reduced to make room for more details on Adoptive cellular therapy.
- in the Introduction chapter:
--implement the description of cellular therapies bases from a general point of view without many biological details which are reported in the further chapters of this review.
-- Introduce recent guidelines for advance HCC treatment on Atezolizumab + Bevacizumab ( Not only Nivolumab which is not in first line treatment).
– FIGURE 1a and 1b:
The Figure 1a, is not innovative and unnecessary, because not centered on Cellular therapy. It could be changed into a Figure/Scheme 1a on Systemic and Immune-based therapies for HCC, including TKIs, Check point inhibitors, Cellular therapies,“Non -Gene modified adoptive therapies” and “gene modified adoptive therapies”.
Therapies not mentioned in the text should not mentioned in the Figures (i.e. apatinib, tremelilumab…). More over many therapies mentioned in this Figure 1b are not approved for advanced HCC and are still under evaluation, while Atezolizumab (approved) is not mentioned, please take in account of it when you draw the new picture. To differentiate what is it approved and what is under evaluation could be necessary.
So we can imagine a Figure 1b “Mechanism of Immune-based therapies for HCC” which explain the different biological mechanism of cellular therapies.
Mention the Figure properly in text.
- In the text (pages 4 to 8) is firstly described the CAR-T cells and successively the TCR-T cells system, the authors should maintain this order also in the Figures: Figure 2a: CAR-T cells model, Figure 2b: CAR-T evolving over time, Figure 2c: TCR-T cell model.
Figures 2a and 2c could be amplified with a target model and type of immune action (for instance the MHC dependent or independent manner).
The text of the Figures2 should be mildly synthesized.
- Chapter 4: What is the Future of ACT for HCC:
A new Figure could help in explaining some of the concepts described in this chapter and would be helpful (i.e. the concepts of physical barrier, HCC TME, chemokines environment, target antigens...)
- From the graphical point of view, the division in sub-chapters should be more evident, for instance (line 147) Chimeric AntigenRecepetor T-calls ( CAR-T) as Line 315 T-cell Receptor transduced T cells (TCR-T) should be more in evidence (i.e. uppercase Or bulleted list)
- 144 references are too many and make their reading difficult.
- line 321 HLA-A0201 is written differently from line 343.
-line 341: iCas9 please explain.
-line 425 please explain “HTS-IR”
- 447 “Modest benefit” is to reductive for the new actual HCC therapy.
Author Response
R1 COMMENT 1: In Simple summer or in Abstract, the sentences on the epidemiology of HCC can be reduced to make room for more details on Adoptive cellular therapy.
AUTHOR RESPONSE: We have added a sentence to both the simple summary and the abstract regarding CAR-T and TCR-T therapeutics, which are further elaborated on in the main body of the review.
R1 COMMENT 2: In the Introduction chapter implement the description of cellular therapies bases from a general point of view without many biological details which are reported in the further chapters of this review.
AUTHOR RESPONSE: We agree with the reviewer and have introduced the following text to address this comment: ‘Adoptive cellular therapy (ACT) using both gene modified and non-gene modified T-cells as a therapeutic modality in HCC is gaining traction. Redirecting T cells to recognize HCC tumour associated antigens is now possible using gene engineering approaches to induce programmed expression of synthetic cell surface receptors on T cells. and Examples include chimeric antigen receptor T-cells (CAR-T), and T-cell receptor modified T-cells (TCR-T) therapies, which are now being tested in clinical trials.’
R1 COMMENT 3: In the introduction, introduce recent guidelines for advance HCC treatment on Atezolizumab + Bevacizumab ( Not only Nivolumab which is not in first line treatment).
AUTHOR RESPONSE: We agree with the reviewer and have added this information to both the text and to Figure 1.
R1 COMMENT 4: The Figure 1a, is not innovative and unnecessary, because not centered on Cellular therapy. It could be changed into a Figure/Scheme 1a on Systemic and Immune-based therapies for HCC, including TKIs, Check point inhibitors, Cellular therapies,“Non -Gene modified adoptive therapies” and “gene modified adoptive therapies”. Therapies not mentioned in the text should not mentioned in the Figures (i.e. apatinib, tremelilumab…). More over many therapies mentioned in this Figure 1b are not approved for advanced HCC and are still under evaluation, while Atezolizumab (approved) is not mentioned, please take in account of it when you draw the new picture. To differentiate what is it approved and what is under evaluation could be necessary.
So we can imagine a Figure 1b “Mechanism of Immune-based therapies for HCC” which explain the different biological mechanism of cellular therapies. Mention the Figure properly in text..
AUTHOR RESPONSE: We have adjusted Figure 1 in line with the reviewer’s suggestions to be more comprehensive and to delineate clearly between approved and non-approved therapeutics.
R1 COMMENT 5: In the text (pages 4 to 8) is firstly described the CAR-T cells and successively the TCR-T cells system, the authors should maintain this order also in the Figures: Figure 2a: CAR-T cells model, Figure 2b: CAR-T evolving over time, Figure 2c: TCR-T cell model.
AUTHOR RESPONSE: We have adjusted Figure 2 in line with the reviewer’s comments.
R1 COMMENT 6: Figures 2a and 2c could be amplified with a target model and type of immune action (for instance the MHC dependent or independent manner). The text of the Figures2 should be mildly synthesized.
AUTHOR RESPONSE: We have introduced Table 3 to address the reviewer’s comment, to outline the broad differences/mechanisms between the cellular therapies described in the main body of the text.
R1 COMMENT 7: Chapter 4: What is the Future of ACT for HCC: A new Figure could help in explaining some of the concepts described in this chapter and would be helpful (i.e. the concepts of physical barrier, HCC TME, chemokines environment, target antigens...)
AUTHOR RESPONSE: We agree that this would be most helpful and have reconfigured Figure 3 to address this suggestion.
R1 COMMENT 8: From the graphical point of view, the division in sub-chapters should be more evident, for instance (line 147) Chimeric AntigenRecepetor T-calls ( CAR-T) as Line 315 T-cell Receptor transduced T cells (TCR-T) should be more in evidence (i.e. uppercase Or bulleted list)
AUTHOR RESPONSE: We agree that this would be helpful and have made this more obvious by the use of bold and italic text.
R1 COMMENT 9: 144 references are too many and make their reading difficult.
AUTHOR RESPONSE: We have made some efforts to reduce the number of references, but there is a great deal of preclinical and clinical research summarized in this review and it is impossible to reduce the references further without losing the breadth of coverage.
R1 COMMENT 10: Line 321 HLA-A0201 is written differently from line 343.
AUTHOR RESPONSE: We have addressed this.
R1 COMMENT 11: Line 341: iCas9 please explain.
AUTHOR RESPONSE: We have addressed this.
R1 COMMENT 12: Line 425 please explain “HTS-IR”
AUTHOR RESPONSE: We have addressed this.
Reviewer 2 Report
In this review manuscript, Roddy et al. review the landscape of cellular immunotherapy for HCC, defining antigenic targets, outlining the range of cell therapy products being applied in HCC, and exploring how advanced engineering solutions may further enhance this therapeutic approach. Adoptive cellular therapy for solid tumors is in its infancy and is challenging for a variety of reasons, outlined in this review. To date, cellular immunotherapy for HCC has demonstrated modest benefits. Combinatorial approaches targeting several aspects of HCC biology concurrently are likely to improve ACT for HCC and to improve clinical outcomes. This review article is generally clear about the current relevant knowledge of the review. However, some questions need to be addressed clearer before the manuscript is suitable for publication.
Major comments
- In the Introduction section, add atezolizumab plus bevacizumab have become the standard of care in first-line therapies for advanced HCCA. (Finn, R. S. et al. Atezolizumab plus bevacizumab in unresectable hepatocellular carcinoma. N. Engl. J. Med. 382, 1894–1905)
- Figure1, 2, and table 1 were not uploaded in the manuscript. I can't see the figure and table.
Author Response
REVIEWER 2 (R2):
R2 COMMENT 1: In the Introduction section, add atezolizumab plus bevacizumab have become the standard of care in first-line therapies for advanced HCCA. (Finn, R. S. et al. Atezolizumab plus bevacizumab in unresectable hepatocellular carcinoma. N. Engl. J. Med. 382, 1894–1905)
AUTHOR RESPONSE: We agree with the reviewer and have added this to the introduction.
R2 COMMENT 2: 2. Figure1, 2, and table 1 were not uploaded in the manuscript. I can't see the figure and table.
AUTHOR RESPONSE: We have reconfigured the figures and have added an extra table such that the updated review contains 3 figures and 3 tables.
Reviewer 3 Report
This is an excellent review article on an emerging topic.
I would suggest the following:
1.The article would be more attractive with a figure and possibly also with a table summarizing the approaches (particularly the MoA for the cell therapies with respect to the TME).
2.I found the authors' personal opinion on the future of HCC therapy not so clear. How would you stratify patients for different cell therapies? Would this be in combination to ICI or sequential? The authors may elaborate on this topic, possibly also with a graphical sketch summarizing different directions / options.
3.It is of utmost important to consider the severity of fibrosis/cirrhosis when designing new therapies against HCC. Would any of the cell therapies be feasible in cirrhosis? What would be their impact on inflammation in cirrhosis?
4.The authors may briefly comment on biomarkers for chosing the rigth cell therapy and monitoring responses.
Author Response
REVIEWER 3 (R3):
R3 COMMENT 1: The article would be more attractive with a figure and possibly also with a table summarizing the approaches (particularly the MoA for the cell therapies with respect to the TME).
AUTHOR RESPONSE: We agree with the reviewer and have added Table 3 summarizing key differences between these cellular therapeutics. We have also amended Figure 3 to include more detail on next generation approaches to overcome limitations associated with cellular therapy for HCC, including details on the TME.
R3 COMMENT 2: I found the authors' personal opinion on the future of HCC therapy not so clear. How would you stratify patients for different cell therapies? Would this be in combination to ICI or sequential? The authors may elaborate on this topic, possibly also with a graphical sketch summarizing different directions / options.
AUTHOR RESPONSE: We agree that more clarity is required and have added the following text to address the reviewer’s comment: ‘ACT for solid tumours is in its infancy and is challenging for a variety of reasons, outlined in this review. HCC remains a strong candidate for cellular immunotherapy, due to its underlying immune and inflammatory pathogenesis, and progress to date in the field has been very encouraging. Drawing from learnings in the blood cancer space and with a deeper understanding of HCC pathophysiology, current investigational T cell therapies may be improved by modifications to enhance HCC targeting, trafficking and immune-resistance, towards improved safety and clinical outcomes. To this end, further preclinical refinements and patient access to early phase clinical trials is key. Currently, cell therapies for HCC are at early stage of development and none have yet been approved. Their position in the therapeutic algorithm will be determined by the magnitude of benefit demonstrated in large clinical trials. Ultimately, rational combinations and/or sequencing of T-cell therapies with small molecules and CPI may be required for optimal clinical outcomes in advanced HCC and should be explored in well-designed clinical studies.’
R3 COMMENT 3: It is of utmost important to consider the severity of fibrosis/cirrhosis when designing new therapies against HCC. Would any of the cell therapies be feasible in cirrhosis? What would be their impact on inflammation in cirrhosis?
AUTHOR RESPONSE: We agree that this is an important issue in developing novel therapies for HCC. We have added the following text to address this reviewer’s comment: ‘To date, approved drugs for advanced HCC have been evaluated in Child Pugh A liver disease in patients with preserved liver function. We strongly recommend that clinical trials of cellular therapies in HCC apply the same criteria, such that potential hepatotoxicity associated with these novel therapies can be appropriately defined and managed.’
R3 COMMENT 4: The authors may briefly comment on biomarkers for chosing the rigth cell therapy and monitoring responses
AUTHOR RESPONSE: We thank the reviewer for this suggestion and have added the following text to address this: ‘In the future, it is possible that novel predictive biomarkers will inform treatment selection and sequencing of drugs for HCC. Presently, the only biomarkers for eligibility in CAR-T and TCR-T trials are target antigen expression and (for TCR-T only) HLA expression analysis. It is too early to predict whether biomarkers for response and toxicity will be identified using ACT for HCC, but current clinical trials will be informative in this regard.’
Round 2
Reviewer 2 Report
I have no other comments. This article is well written and acceptable.